# STRIPAK Dependent and Independent Phosphorylation of the SIN Kinase DBF2 Controls Fruiting Body Development and Cytokinesis during Septation and Ascospore Formation in *Sordaria macrospora*

**DOI:** 10.3390/jof10030177

**Published:** 2024-02-26

**Authors:** Maria Shariatnasery, Valentina Stein, Ines Teichert, Ulrich Kück

**Affiliations:** 1Allgemeine und Molekulare Botanik, Fakultät für Biologie und Biotechnologie, Ruhr-University Bochum, Universitätsstr. 150, NI 06/280, D-44780 Bochum, Germany; maria.shariatnasery@ruhr-uni-bochum.de (M.S.); valentina.stein@ruhr-uni-bochum.de (V.S.); ines.teichert@uni-goettingen.de (I.T.); 2Abteilung Forstbotanik und Baumphysiologie, Büsgen-Institut, Georg-August-Universität Göttingen, Büsgenweg 2, D-37077 Göttingen, Germany

**Keywords:** SIN kinase DBF2, STRIPAK, fungal development, *Sordaria macrospora*

## Abstract

The supramolecular striatin-interacting phosphatases and kinases (STRIPAK) complex is highly conserved in eukaryotes and controls diverse developmental processes in fungi. STRIPAK is genetically and physically linked to the Hippo-related septation initiation network (SIN), which signals through a chain of three kinases, including the terminal nuclear Dbf2-related (NDR) family kinase DBF2. Here, we provide evidence for the function of DBF2 during sexual development and vegetative growth of the homothallic ascomycetous model fungus *Sordaria macrospora*. Using mutants with a deleted *dbf2* gene and complemented strains carrying different variants of *dbf2*, we demonstrate that *dbf2* is essential for fruiting body formation, as well as septum formation of vegetative hyphae. Furthermore, we constructed *dbf2* mutants carrying phospho-mimetic and phospho-deficient codons for two conserved phosphorylation sites. Growth tests of the phosphorylation mutants showed that coordinated phosphorylation is crucial for controlling vegetative growth rates and maintaining proper septum distances. Finally, we investigated the function of DBF2 by overexpressing the *dbf2* gene. The corresponding transformants showed disturbed cytokinesis during ascospore formation. Thus, regulated phosphorylation of DBF2 and precise expression of the *dbf2* gene are essential for accurate septation in vegetative hyphae and coordinated cell division during septation and sexual spore formation.

## 1. Introduction

The striatin-interacting phosphatase and kinase complex (STRIPAK) is a highly conserved eukaryotic protein complex that contains the heterotrimeric phosphatase protein phosphatase 2A (PP2A) with its regulatory B‴ subunit striatin [1,2,3,4]. This highly conserved eukaryotic protein complex is central to diverse developmental processes. For example, in lower eukaryotes such as fungi, sexual and asexual development is disturbed in mutants lacking genes for STRIPAK subunits [5,6,7,8,9], and in humans, different cancer types are linked to defects in subunits of the STRIPAK complex [10,11,12]. 

A substantial number of studies in diverse systems have shown that STRIPAK plays an active role in regulating the phosphorylation of diverse proteins, some of which are members of associated signalling pathways, including the Hippo pathway or MAPK (mitogen-activated protein kinase) cascades [3,10,13]. We recently applied absolute quantification of protein phosphorylation by parallel-reaction monitoring (PRM) to demonstrate STRIPAK-dependent phosphorylation of the Hippo-like germinal centre kinase (GCK) SmKIN3 [13]. This kinase is a component of the fungal septation initiation network (SIN), a Hippo-related signalling pathway [14,15].

Here, we describe a functional analysis of the NDR (Nuclear Dbf2-related) kinase DBF2, the final kinase of the three-tiered SIN cascade [16,17]. DBF2 is a member of a highly conserved family of kinases found across eukaryotes. NDR kinases play a vital regulatory role in several cellular processes, including centrosome duplication, cytokinesis, mitotic exit, morphogenesis, and cell growth and development from yeast to metazoans [18,19,20,21]. DBF2 homologues have been investigated in a wide range of ascomycetes. For example, baker’s yeast *Saccharomyces cerevisiae* contains three essential components that share structural and functional similarities with the Hippo pathway, namely the NDR kinase DBF2, the coactivator MOB1, and the Ste20-like kinase CDC15. DBF2 in yeast has been identified as part of the CCR4 transcriptional complex, which is responsible for regulating many genes. This complex has both positive and negative effects on gene expression [22,23]. Moreover, disruption of DBF2 in yeast leads to glycogen accumulation, indicating its involvement in glycogen metabolism [24]. In the human pathogenic yeast *Candida albicans*, the DBF2 homologue is essential for cell viability [25]. In filamentous ascomycetes, SIN components control a range of complex and unique cellular processes, including hyphal elongation and conidiation. In *Aspergillus nidulans* for example, the DBF2 orthologue SIDB has acquired additional functions beyond its conserved role in cell cycle control and cytokinesis. Notably, it is involved in regulating asexual reproduction, since disruption of SIDB leads to the complete loss of conidiation [26,27]. Furthermore, in *Neurospora crassa*, DBF-2 is involved in cell cycle regulation, glycogen biosynthesis, and conidiation [28].

In this study, we used *Sordaria macrospora* as an experimental system [29] to study the function of DBF2 in vivo. This homothallic fungus is an ideal system to decipher components that are involved in sexual development and propagation. The DBF2 kinase from *S. macrospora* is homologous to LATS1, the terminal kinase of the Hippo pathway in animals. The Hippo pathway is known for its crucial roles in regulating cell proliferation and promoting apoptosis during cellular differentiation. Overexpression of the LATS1 gene induces G2-M phase arrest and reduces the ability of breast cancer cells to form tumours [30,31,32]. The growing evidence highlighting the role of Hippo signalling in cancer biology indicates that targeting this pathway could provide new possibilities for alternative therapeutics [33].

The above findings prompted us to thoroughly investigate DBF2 functions including its conserved phosphorylation sites in *S. macrospora*. We show that lack of DBF2 severely affects sexual development, including the formation of ascospores. We provide evidence that conserved amino acid sites in DBF2, which are phosphorylated in a STRIPAK dependent or independent manner, are relevant for proper hyphal septation, stress response, and formation of sexual spores. Finally, we found that overexpression of *dbf2* substantially affects cytokinesis during ascospore formation, indicating that this process is also SIN-dependent. To the best of our knowledge, this is the first report on how the terminal kinase of the SIN pathway affects and regulates sexual development in euascomycetes.

## 2. Material and Methods

### 2.1. Strains and Growth Conditions

Electro-competent *E. coli* cells XL1 Blue MRF’ were used for cloning and propagation of recombinant plasmids under standard laboratory conditions [34,35]. All *S. macrospora* strains used in this work are listed in Appendix A and were grown under standard conditions [36]. Isogenic and homokaryotic strains were generated by genetic crossing and ascospore isolation [37]. To obtain phospho-mutants, the mutated plasmids (Appendix A) were transformed into a ∆dbf2 strain. Phospho-mutations in the generated strains were verified by PCR analysis and DNA sequencing (Eurofins Genomics; Ebersberg, Germany).

### 2.2. Plasmid Construction and Generation of Deletion Strains

Plasmids were generated through different methods. Firstly, PCR or restriction fragments were cloned into vector backbones by using T4 DNA ligase in a process known as restriction and ligation. Alternatively, Golden Gate cloning, as described previously, was used for constructing deletion plasmids. Additionally, some plasmids were constructed using homologous recombination in yeast [38]. For site-directed mutagenesis, the Q5^®^ Site-Directed Mutagenesis Kit from New England Biolabs was employed, following the supplier’s protocol. All plasmids used for *S. macrospora* transformations are listed in Appendix A, and oligonucleotides, used for PCR, DNA sequencing, and in vitro mutagenesis in this study are listed in Appendix A. Deletion strains were generated using a *Bsa*I-mediated Golden Gate cloning system [39] (Appendix A).

### 2.3. Sequencing of DNA 

Plasmid DNA sequencing was conducted either by Eurofins Genomics located in Ebersberg, Germany or by the Department for Biochemistry at Ruhr-University Bochum (both Sanger sequencing technology). The sequencing data obtained were subsequently analysed using SnapGene software (GSL Biotech LLC, San Diego, CA, USA, Version 7.1.0).

### 2.4. Genetic Crosses of S. macrospora 

To generate homokaryotic strains with different dbf2 versions, transgenic strains possessing the normal spore colour (black) were crossed with a strain carrying the spore colour mutation *fus1-1*, resulting in brown ascospores. Following cultivation on cornmeal medium (BMM) for a period of 7–9 days, perithecia that contained spores exhibiting two distinct colours within a single ascus were isolated. The ascospores from these perithecia were then inoculated onto BMM-NaAc medium to induce germination. To ensure the successful isolation of the desired strain, the germinated spores were tested for specific resistance or phenotypic characteristics.

### 2.5. Growth and Stress Test 

Using three technical replicates, growth and stress tests were conducted three times for strains, as mentioned in Section 3. After pre-culturing the strains on BMM medium for a period of 4 days, a standardized piece of the pre-culture was used to inoculate SWG plates that contained 20 mL of SWG medium [40]. Mycelial growth was measured at two time points: 48 and 72 h following inoculation. For cell wall stress assays, the SWG medium was supplemented with 0.01% (*v*/*v*) SDS plus 2 mg/mL Congo Red.

### 2.6. Quantification of Fruiting Body Formation 

Fruiting body initiation was measured by quantifying development after a three-day incubation period on BMM-covered microscope slides. To assess fruiting body formation, pictures of the strains were captured after seven days of incubation in Petri dishes containing solid BMM. Imaging was conducted using a Zeiss Stemi 2000-C binocular microscope equipped with an AxioCam ERc5s camera and Zen 2 core (version 2.5, Zeiss, Oberkochen, Germany).

### 2.7. Microscopic Investigation

Conventional microscopy in this study was conducted using the Axioimager.M1 microscope from Zeiss, equipped with Metamorph 7.7 software, following the procedures outlined previously [41]. For fluorescent imaging, a SPECTRA X 6 LCR SA LED lamp from Lumencor was used for fluorophore excitation. The Chroma filter set 31000v2 (comprising a D350/50 excitation filter, D460/50 emission filter, and 400dclp beam splitter) was used to detect DAPI and calcofluor-white (CFW) staining. EGFP was detected using the filter set 49002, which includes an HQ470/40 excitation filter, HQ525/50 emission filter, and T495LPXR beam splitter. Imaging was performed using a Photometrix Cool SnapHQ camera from Roper Scientific (Martinsried, Germany). The acquired images were converted using Metamorph software and processed using Adobe Photoshop CS6. 

To investigate sexual development and perform fluorescent imaging, the strains were inoculated on slides covered with BMM and then mounted with a coverslip using a 0.7% NaCl solution. The initiation of fruiting body development was quantified after three to four days of incubation on BMM-covered microscope slides. For assessment of fruiting body formation, pictures of strains were taken after seven to ten days of incubation in Petri dishes on solid BMM (with a Zeiss Stemi 2000-C binocular, using an AxioCam ERc5s with the software Zen 2 core (version 2.5, Zeiss). These experiments were performed for three biological replicates with three technical replicates each [37]. 

For fluorescent imaging of hyphae, imaging was conducted after two days of incubation. To assess septa in the vegetative mycelium and ascogonial coils, the cell wall was stained using CFW (Sigma Aldrich, St. Louis, MO, USA). A stock solution of CFW at a concentration of 1 μg/mL was prepared, and then diluted 1:400 in a 0.7% NaCl solution for staining. Hyphal fusion events were investigated by observing strains that were grown on a layer of cellophane (Bio-Rad, Munich, Germany) placed on top of solid minimal medium (MMS) containing starch for thinner mycelial growth for a duration of two days [42,43]. The septum distances were examined after 2 days of incubation on BMM. On each of three independent BMM-coated glass slides, at least 150–200 septum distances were examined from the hyphal tip to the colony centre.

### 2.8. Isolation of Proteins from S. macrospora

For testing protein expression in transformants, the standard protein extraction method for filamentous fungi was used as described [44]. Strains were grown at 27 °C in a Petri dish with liquid BMM for three days. For cell wall lysis and protein extraction, mycelium was harvested and frozen in liquid nitrogen. The frozen mycelium was ground in liquid nitrogen and suspended in FLAG extraction buffer (50 mM Tris-HCl pH 7.4, 250 mM NaCl, 10% (*v*/*v*) glycerol, 0.05% (*v*/*v*) NP-40, 1 mM PMSF, 0.2% (*v*/*v*) protease inhibitor cocktail IV, 1.3 mM benzamidine, 1% (*v*/*v*) phosphatase inhibitor cocktails II and III). After centrifugation at 4 °C and 15,000 rpm for 30 min, the supernatant was used for SDS-PAGE.

### 2.9. Western Blot and Immunodetection

Protein concentrations were determined using the Bradford assay [45]. SDS-PAGE was performed in a vertical electrophoresis system MiniPROTEAN^®^ Tetra Cell (Bio-Rad, Hercules, CA, USA) with a 5% stacking gel and 12% separation gel [46]. For each sample, 15 µg of protein extract from each fungal transformant was diluted in 16 µL distilled water, plus 4 µL of 5 × SDS sample buffer, and incubated at 95 °C for 10 min. The prestained PageRuler Protein Ladder (Thermo Scientific, Waltham, MA, USA) served as a protein size standard. Gels with separated proteins were stained by Coomassie brilliant blue solution followed by de-staining with 10% (*v*/*v*) acetic acid. Western blot analysis was performed on nitrocellulose membranes. Anti-GFP Monoclonal Antibody (JL-8) (TaKaRa Bio Europe/Clontech, Saint-Germain-en-Laye, France), and HRP-linked anti-mouse IgG antibody (Cell Signaling Technology; Frankfurt am Main, Germany) were used as the primary and secondary antibodies. Alpha-Tubulin antibodies were obtained from Oncogene Company (now EMD Millipore). The SuperSignal™ West Pico PLUS Chemiluminescent Substrate (Thermo Scientific; Waltham, MA, USA) was used as the signal detection reagent, and imaging was performed with the Chemidoc XRS+ System (Bio-Rad; Hercules, CA, USA), with an exposure time of 20–80 s.

## 3. Results

### 3.1. Construction of dbf2 Deletion Strains for Functional Analysis

For a functional characterization of DBF2, a *dbf2* deletion strain (Δdbf2) was constructed as described above in Section 2. The *dbf2* gene was substituted by a hygromycin B resistance cassette using mutant Δku70, a non-homologous end-joining-deficient strain [29]. To generate Δdbf2 (Appendix A), linearized plasmid pMSD16-dbf2 was transformed into Δku70. The primary transformants were tested for hygromycin B resistance and were analysed using PCR. Six primary transformants for *dbf2* were obtained and subsequently crossed to fus1 to obtain homokaryotic strains carrying solely the ∆dbf2 mutation, but not the wild type *dbf2.* An ascospore isolate was selected as the ∆dbf2 strain, which was used as the recipient strain for further analysis.

### 3.2. The DBF2 Kinase Affects Hyphal Septation and Fruiting Body Formation

Mycelia from wild type and ∆dbf2 were grown on BMM-coated glass slides for 48 h, and then stained with CFW for microscopic investigation of cell walls. Septa were produced evenly across leading and branching hyphae in the wild type (Figure 1). In contrast, Δdbf2 completely lacked septa even after prolonged vegetative growth. To confirm that the observed defect is due to the deletion of *dbf2*, we performed a complementation analysis. Two different complementation constructs were generated to express the *dbf2* gene. In the pNAdbf2 construct, *dbf2* gene expression is controlled by its native promoter. In a comparative RNA-Seq study, we found previously that dbf2 expression is comparatively weak [47]. Therefore, in the second construct, pOEdbf2, the constitutive and strong *gpd* promoter from *A. nidulans* controls the transcriptional expression of *dbf2* [48,49]. To conduct the complementation analysis, the two plasmids, pOE*dbf2* and pNA*dbf2*, both carrying a nourseothricin (*nat*) resistance cassette as a selectable marker, were introduced into Δ*dbf2*. Primary transformants were selected based on their resistance to hygromycin B and nourseothricin. Following the selection process, ascospore isolates were generated from the fertile transformants, and these isolates were subjected to phenotypic analysis. Data from representative isolates carrying pNA*dbf2* (∆dbf2::*NAdbf2*) and pOE*dbf2* (∆dbf2::*OEdbf2*) demonstrated that complementation constructs were effective in restoring the wild type-like septation pattern in leading hyphae (Figure 1). 

Sexual development and hyphal fusion of Δdbf2 were analysed and compared to the wild type and complementation stains. Ascogonia formed within three days, and later developed into unpigmented and subsequently pigmented protoperithecia after five days (Figure 2). After seven days, fully developed and mature perithecia were observed. In contrast, the strains lacking the *dbf2* gene displayed significant deficiencies in fruiting body formation. They produced only small pigmented protoperithecia that were unable to differentiate further. 

Sexual development was successfully restored in all complemented strains Δdbf2::*NAdbf2*, Δdbf2::*OEdbf2*, as well as in Δdbf2::*OEdbf2-gfp*. Interestingly, only strain Δdbf2::*NAdbf2* generated asci carrying eight ascospores. Where *dbf2* was under the transcriptional control of the gpd promoter (Δdbf2::*OEdbf2*, Δdbf2::*OEdbf2-gfp)*, we observed huge ascospores that were sausage-shaped (Figure 2). As described below, these ascospores were further investigated by fluorescence microscopy and DAPI staining.

### 3.3. Phospho-Mimetic and Phospho-Deficient dbf2 Mutants Show a Septation Phenotype and Altered Stress Responses

Our recent comprehensive global proteome and phosphoproteome analysis [13,50] provided valuable insights into potential dephosphorylation targets of STRIPAK. In these studies, protein samples from wild type and three STRIPAK mutants were used for absolute quantification through parallel reaction monitoring (PRM). This investigation analysed phosphorylation site occupancy in components of the septation initiation network (SIN), including DBF2, identifying three phosphorylation sites (S89, S104, S502) in DBF2 [50]. While phosphorylation of S104 was STRIPAK dependent, phosphorylation of S89 and S502 were STRIPAK independent [13]. Notably, the phosphorylation site S104 in DBF2 exhibited significantly decreased phosphorylation levels when the STRIPAK complex was non-functional, suggesting its dependency on STRIPAK. Conversely, the phosphorylation site S502, which is investigated in this study, did not exhibit any differential regulation in STRIPAK deletion mutants.

The *S. macrospora dbf2* gene (SMAC_05230) comprises an open reading frame (ORF) of 2217 bp. The predicted DBF2 protein has a length of 704 amino acids and shows high identity to homologues in *N. crassa* (96%), *A. nidulans* (61%), and *S. pombe* (53%) (Figure 3A). The sequence comparison between seven DBF2-like proteins from protoascomycetes and euascomycetes (Figure 3B) shows that both phosphorylated residues S104 and S502 are highly conserved. However, S104 is contained in a region of DBF2 that is less conserved in the two yeasts investigated. 

To understand the effects of these phosphorylation sites on sexual development and hyphal growth, we generated different versions of *dbf2* with phospho-mimetic or -deficient modifications. We substituted single base pairs in the triplets encoding S104 and S502, generating codons for alanine (no phosphorylation mimicry) or glutamic acid (mimics phosphorylation due to its negative charge) (Appendix A). We used PCR-directed mutagenesis to generate plasmids that express *dbf2*-S104A or -S104E, and *dbf2*-S502A or -S502E gene variants. The variants were inserted into plasmid pMSC8, which carries the *dbf2* gene fused to *gfp*. For DNA-mediated transformation of recipient strain ∆dbf2, we used plasmids carrying phospho-mimetic (pOE*dbf2*-S104E and pOE*dbf2*-S502E) and phospho-deficient (pOE*dbf2*-S104A, pOE*dbf2*-S502A) variants of *dbf2*. 

Complementing the deleted *dbf2* gene with the desired variants allowed us to evaluate their impact on various cellular processes. Primary transformants were identified based on their resistance to hygromycin B and nourseothricin. Three homokaryotic ascospore isolates obtained from these fertile transformants were examined for each phospho-mimetic and phospho-deficient strain. The expression of wild type DBF2-GFP and the phospho-variants was confirmed through Western blot analysis, demonstrating the synthesis of a predicted 107-kDa protein (Figure 4). Subsequently, we conducted phenotypic analyses on individual isolates that carried *dbf2*-S104A, *dbf2*-S104E, *dbf2*-S502A, or *dbf2*-S502E. Complementation with the wild type *dbf2-gfp* gene resulted in generating fully fertile strains exhibiting normal mycelial growth, indicating the full functionality of the fusion gene (Figure 5A). 

The individual isolates were compared with wild type, ∆dbf2, and complemented transformants (Δdbf2::NAdbf2 and Δdbf2::OEdbf2-*gfp*) for sexual development and hyphal growth. Analysis of various developmental stages showed that within seven days, both the phospho-mimetic (S104E, S502E) and phospho-deficient dbf2 mutants (S104A, S502A) formed ascogonia, protoperithecia, and pear-shaped perithecia (Figure 5A). However, compared to wild type and complemented strains carrying the wild type gene, we observed a notable reduction in the number of asci and ascospores generated within perithecia of all phospho-mutants. After 10 days of incubation, a decrease in the number of asci and ascospores was observed for all phospho-mutants in comparison to the wild type and completed strains. Specifically, for S104A, no discharged ascospores were observed, while S104E and S502A exhibited very few, and S502E had a slightly higher number of ascospores (Figure 5B).

In Δ*dbf2*::*OEdbf2*-S502A and Δ*dbf2*::*OEdbf2*-S502E, we observed only sausage-shaped ascospores, which resembles the complemented strains described above expressing *dbf2* from the strong *gpd* promoter (Δdbf2::*OEdbf2*-*gfp*). However, in Δ*dbf2*::*OEdbf2*-S104A and Δ*dbf2*::*OEdbf2*-S104E, a mixture of wild type-like and sausage-shaped ascospores was observed (Figure 5B). Next, we examined the hyphal fusion phenotype in diverse mutant strains by examining at least three individual samples for each strain. The mycelia were grown for two days, except for ∆dbf2, which had to be grown for at least four days. All the phospho-mimetic and phospho-deficient strains tested demonstrated the ability to undergo hyphal fusion, which was not distinguishable from the wild type or complemented strains (Figure 5C). Hyphal fusion events were even detected in the deletion strain, although they showed significantly reduced growth rates. 

Analysing vegetative growth on SWG medium, we noted that Δ*dbf2*::*NAdbf2*, Δ*dbf2*::*OEdbf2-gfp*, and Δ*dbf2*::*OEdbf2*-S104E had similar growth rates as the wild type (Figure 6). In contrast, Δdbf2 had a significantly decreased growth rate compared to the wild type, and Δ*dbf2*::*OEdbf2*-S104A, Δ*dbf2*::*OEdbf2*-S502A and Δ*dbf2*::*OEdbf2*-S502E had a partial reduction in growth compared to Δdbf2. 

The response to cell wall stress was evaluated by adding Congo Red to the SWG medium, following the same procedure as the vegetative growth assay. The growth rate was observed for 3 days, except Δdbf2, which was investigated for 12 days. The wild type as a control strain, displayed a growth of 2.0 ± 0.0924 cm/d on the cell wall stress medium (Figure 6); however, the growth of Δdbf2 ceased at a size of 2–3 mm after 7 days, and no further growth was observed throughout the duration of a 12-day experiment. The complemented strains carrying the wild type *dbf2* gene showed no sensitivity to cell wall stress. Therefore, the impaired adaptation to cell wall stress seen in Δ*dbf2* was fully restored by an ectopically expressed *dbf2* gene. However, phospho-mutants exhibited differences compared to the wild type. Pronounced sensitivity to cell wall perturbing agents was observed in three of the phospho-deficient mutants, with the exception of phospho-mimetic mutant Δ*dbf2*::*OEdbf2*-S502E exhibiting a tolerance response to cell wall perturbing agents comparable to the control (Figure 6).

Since Δdbf2 showed a severe septation phenotype, we further analysed septation in the strains carrying phospho-mimetic and phospho-deficient variants. Our comparative results showed that the phospho-DBF2 mutants exhibit an increased distance between septa (Figure 7A). To quantify our observation, we performed CFW staining and examined a minimum of 150 septum distances for each strain in three technical replicates, meaning at least 450 septum distances were measured for each strain. Our quantitative analysis indicated that both phospho-mimetic and phospho-deficient DBF2 mutations significantly affected distances between septa: compared to the wild type, the S104 mutations showed an almost one-fold increase in distances between septa, while the S502 phosphorylation site mutations resulted in an approximately two-fold increase. When we introduced the wild type *dbf2* gene into the deletion strain, we observed septum distances comparable to the wild type strain (Figure 7B). These findings indicate that the mutations affecting DBF2 phosphorylation have a significant impact on septum formation and thus also on cytokinesis [51].

### 3.4. Overexpression of dbf2 Results in Strains with a Disturbed Cytokinesis

In fungal systems, overexpression of candidate genes is a common experimental approach to examine gene functions. As described above, we complemented Δ*dbf2* with two different constructs carrying *dbf2* expressed from either a native or an overexpression promoter. Complementation with construct pNA*dbf2* resulted in asci and ascospores that appeared indistinguishable from the wild type. In contrast, in all asci the overexpression strains generated a single large spore with a sausage-like shape (Figure 2 and Figure 5B). To investigate nuclear divisions in asci from wild type and *dbf2*-overexpression (∆dbf2::*OEdb*f2*-gfp*) strains in more detail, we performed DAPI staining of asci during different stages of maturation, and here show mature ascospores from wild type and sausage-like ascospores from overexpression strains (Figure 8A), and a schematic drawing of nuclear divisions in wild type asci, representing karyogamy, meiosis I and II, as well as post-meiotic meiosis (Figure 8B).

DAPI staining confirmed that several of these meiotic nuclear division stages are found in both the wild type and ∆dbf2::*OEdb*f2*-gfp* strains. However, the proper formation of ascospores is prevented after post-meiotic mitosis. These data suggest a failure in proper cytokinesis, which prevents the formation of ascospores after meiotic nuclear divisions (Figure 8A,C). Instead, we observed a multinucleate giant ascospore with 8 nuclei within mature asci. Based on these observations, we conclude that dysregulation of the *dbf2* gene is likely responsible for the sausage-shaped phenotype. In summary, in strains with dysregulated *dbf2* gene expression, nuclear division is not affected, but the formation of wild type-like ascospores is. Thus, coordinated *dbf2* gene expression is important for maintaining proper cell division together with coordinated cytokinesis.

## 4. Discussion

In this study, we investigated functions of the NDR kinase DBF2 by constructing *dbf2* mutants carrying phospho-mimetic (serine substituted by glutamate) and phospho-deficient (serine substituted by alanine) codons for two conserved phosphorylation sites (S104 and S502) and performed studies with deleted *dbf2*, overexpressed *dbf2*, and various complementation combinations in strains of the ascomycete *Sordaria macrospora*. We found that altered phosphorylation of the conserved amino acid residues 104 and 502 in DBF2 disturbs hyphal septation and stress response, that coordinated *dbf2* expression controls the cytokinesis of ascospore formation, and that septum formation, hyphal fusion, and fruiting body development are strictly dependent on the SIN kinase DBF2.

The SIN pathway is known to play a vital role in both septum formation and conidiation during diverse fungal developmental stages [52,53]. Removing any of the components of the SIN pathway leads to the formation of aseptate strains in *N. crassa*, *A. nidulans*, *A. fumigatus*, and *S. macrospora* [54,55,56]. The crucial role of the SIN pathway in facilitating septation and cytokinesis in *N. crassa* has been validated through the analysis of SIN mutants. Specifically, the absence of genes encoding CDC-7 and DBF-2 leads to the formation of vegetative mycelium deficient in septation. In contrast, deletion mutants lacking SID-1 and its adaptor CDC-14 produce aseptate mycelium, but this phenotype eventually reverts to the wild type condition [54].

Our findings, along with previous research on SmKIN3 [55], underscore the significance of SIN components for septation in *S. macrospora.* The deletion of *dbf2*, as described here, displayed a significant reduction in vegetative growth, characterized by thinner hyphae and lower density of mycelium compared to the wild type. Intriguingly, deletion of the *dbf2* gene did not affect hyphal fusion. 

Impaired hyphal fusion and sterility are prominent characteristics often observed in parallel in various developmental mutants of *S. macrospora*. For instance, previous studies have shown that all mutants of STRIPAK, including the germinal centre kinase (GCK) deletion mutant ΔSmkin24, exhibited a defect in hyphal fusion, besides being sterile [57]. Previously, we also showed that deletion of *Smkin3*, encoding another GCK, impairs hyphal fusion [55]. The notable association between defects in sexual development and vegetative hyphal fusion (VHF) has given rise to the hypothesis that VHF is a prerequisite for the formation of fruiting bodies. However, recent findings have raised questions about this hypothesis, since mutants have been discovered that are either sterile yet competent in fusion, or fertile but deficient in fusion [58,59]. In these cases, signalling specificity is achieved through various mechanisms, such as scaffold proteins and cross-pathway inhibition, which interact with distinct downstream effectors. Indeed, a considerable number of fungal proteins have been discovered to specifically regulate either VHF or fruiting body formation. For instance, proteins involved in autophagy, including SmATG4, SmATG8, SmJLB1 in *S. macrospora*, and PRM-1 in *N. crassa*, are crucial for the successful completion of the sexual cycle, but not for VHF [59,60]. Notably, our findings in this study indicate that DBF2 may not play a crucial role in the VHF process in *S. macrospora* but seems exclusively involved in sexual development. During sexual development of ascomycetes, the formation of ascogenous croziers (hook-shaped cells) and nuclear assortment prior to meiosis is dependent on septum formation. This process is prevented in sterile Δdbf2 strains, which lack any septum formation. In conclusion, our findings add to the growing list of mutations identified in different organisms that are specific for just one developmental program.

As mentioned in the results, we previously performed quantitative analyses of SIN subunit phosphorylation, generating stable isotope-labelled standard (SIS) peptides to compare their phosphorylation with native peptides isolated from STRIPAK deletion mutants in *S. macrospora* [13,50]. That analysis identified S104 phosphorylation as being STRIPAK dependent [13]. Amino acid S502 in *S. macrospora* is conserved and corresponds to the S499 site in DBF-2 in *N. crassa*. Here, we found that the disordered phosphorylation of both phosphorylation sites significantly affects ascospore development, septum formation, and vegetative hyphal growth. This is consistent with the observation that all the phospho-variants showed a reduced number of mature perithecia, and sexual spores compared to the wild type and deletion strains complemented with the wild type gene. 

The phosphorylation of S104 and S502 appears to have a minor impact on sexual development, since the phospho-mutants were fertile but exhibited only a reduced number of ascospores. Although we observed a minor decrease in vegetative growth rate when the S104 phosphorylation site was deficient, the phospho-mimetic mutation caused no change in growth compared to the wild type. In contrast, for the conserved phosphorylation site S502, both the phospho-deficient and phospho-mimetic mutants exhibited a significant reduction in growth rates. Notably, the phospho-deficient mutants showed a more severe effect on vegetative growth compared to the phospho-mimetic mutants. These findings suggest that proper phosphorylation at these sites plays a crucial role in regulating vegetative growth in *Sordaria macrospora*, with specific phosphorylation events having different impacts on growth outcomes. According to a previous analysis, the de-phosphorylation of SmKIN3, which is controlled by the STRIPAK complex, controls the precise regulation of septum formation and the time-dependent positioning of SmKIN3 on the first septum in a growing hypha. Consequently, this suggests that STRIPAK plays a regulatory role in controlling the de-phosphorylation of SmKIN3 and DBF2, thereby regulating the process of septum formation [13]. Here, the quantitative analysis of septum distances in *dbf2* phospho-mutants confirmed that the abnormal septum formation phenotype is associated with the phospho-mimetic and phospho-deficient DBF2 mutations at both S104 and S502 sites. Importantly, when we introduced the *dbf2* gene without these mutations into the deletion strain, no significant differences in septum distances were observed compared to the wild type strain. 

In contrast to observations in *N. crassa*, where both phospho-deficient and phospho-mimetic variants of S499 failed to restore septation in deletion strains, this study in *S. macrospora* demonstrated that both phospho-mimetic and phospho-deficient variants of S502 were able to restore septation in *dbf2* deletion strains. However, a notable consequence of these substitutions was the presence of significant distances between septa, indicating a reduction in the number of septa formed. These results support the critical role of these two phosphorylation sites in DBF2 during vegetative propagation and septum formation. Furthermore, we compared the data from growth rates of phospho-mutant strains and their respective septum distances. Interestingly, we discovered a negative correlation between the length of septum distances and the vegetative growth rate. When the septum distances were longer, growth rates of the strains were reduced. This observation suggests a potential relationship between the regulation of septum formation and the distance between septa, which could have an impact on the overall growth dynamics of filamentous fungi. 

The results of the cell wall stress test using Congo Red revealed that proper phosphorylation of S104 is crucial for an effective stress response. Specifically, S104 phospho-mimetic and phospho-deficient strains exhibited increased sensitivity to Congo Red, indicating the importance of S104 phosphorylation in mediating the response to the stressor. However, in the case of S502, a different pattern emerged. The phospho-deficient strains displayed increased sensitivity to Congo Red, suggesting that phosphorylation at S502 plays a protective role in response to cell wall stress. In contrast, the phospho-mimetic mutants exhibited enhanced resistance to Congo Red, showing approximately 25% higher resistance compared to the wild type. These findings indicate that mimicking a (constitutive) phosphorylation state of S502 confers increased resistance to cell wall stress. In this context a previous report for *C. albicans* is remarkable, where cell wall stress was shown to affect fungal cytokinesis [61].

Cytokinesis, the final step in cell division, involves the physical separation of a single mother cell into two daughter cells. This process is observed in all eukaryotes, including fungal species and metazoans. Key to cytokinesis is the actomyosin-based contractile ring, which is formed during the onset of mitosis and undergoes constriction after anaphase is completed [62,63,64]. For the cell cycle to be successfully completed, proper coordination between mitosis (cell division) and cytokinesis (cell separation) is necessary in all cells. The SIN pathway plays a critical role in coordinating and controlling the process of cytokinesis in fission yeast cells [65]. Like our results, meiotic divisions appear normal, but SIN mutants cannot form ascospores [66]. We speculate that dysregulation of the *dbf2* gene in *S. macrospora* might result in uncontrolled phosphorylation of a Clp1-homologue, a target of the SIN pathway.

Our observation of huge ascospores with eight nuclei is consistent with findings from *S. pombe*, where mutations in the SIN pathway result in two distinct phenotypes: cells with multiple nuclei or cells with multiple septa that are unable to undergo proper cell cleavage [67]. Probably, the exit from meiosis II with cytokinesis is disturbed, as was observed in diverse fungal mutants [68,69].

In conclusion, our study identified distinct phosphorylation sites in the NDR kinase DBF2 as being essential for stress response, proper septum formation, and ascospore formation in *S. macrospora*. Moreover, DBF2 seems exclusively involved in sexual development and does not contribute to hyphal fusion. Therefore, our findings add to the growing list of developmental factors identified in different organisms specific for just one developmental program. Future work will focus further on SIN pathway phosphorylation linked to cellular development, and thus broaden our understanding of Hippo signalling in animal systems.

## Figures and Tables

**Figure 1 jof-10-00177-f001:**
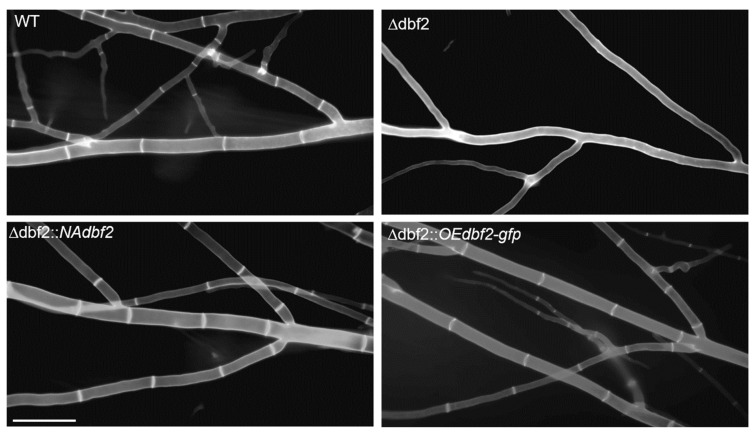
Septation of mycelial hyphae. Images from vegetative cells from wild type and Δdbf2 strains were taken after 48 h of incubation on complete medium (BMM) and stained with calcofluor-white (CFW). The bar indicates 20 µm.

**Figure 2 jof-10-00177-f002:**
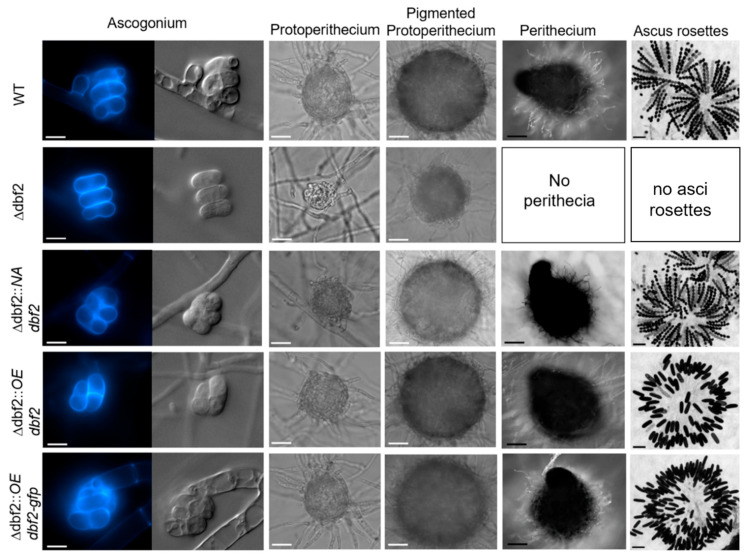
Sexual development of wild type, Δdbf2 and complemented strains. Sexual development of all strains was observed after three (ascogonia), five (unpigmented and pigmented protoperithecia) and seven days (perithecia) on BMM media. Scale bars indicate 20 μm (white) or 100 μm (black). Septation of ascogonia was monitored after staining with CFW.

**Figure 3 jof-10-00177-f003:**
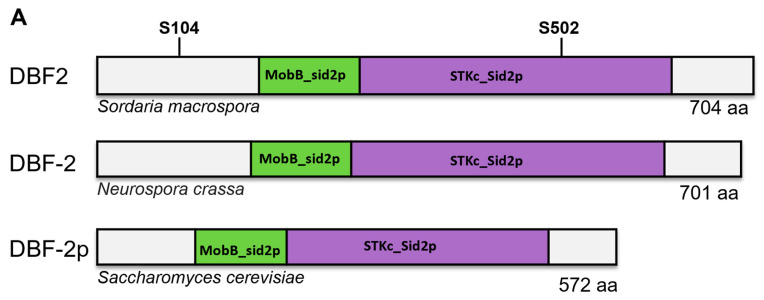
Comparison of DBF2 homologs from ascomycetous fungi. (**A**) Primary structure of DBF2 proteins from *S. macrospora* (S.m) DBF2, N. crassa (N.c) DBF-2, and S. cerevisiae (S.c) DBF-2p. All homologs contain a Mob-binding domain found in fungal Sid2p-like serine/threonine protein kinases (MobB_sid2p) and a serine/threonine-protein kinase (STK_sid2p). Phosphorylation sites of S. macrospora DBF2 are indicated that were investigated in this study. (**B**) Multiple sequence alignment of DBF2 homologs from proto- and euascomycetes. Framed are the conserved phosphorylation residues S104 and S502. Abbreviations: S.m, *Sordaria macrospora* (XP_003347031.1); N.c, *Neurospora crassa* (XP_964888.1); P.a, *Podospora anserina* (CDP24309.1); P.g, *Pyricularia grisea* (XP_030980211.1); A.n, *Aspergillus nidulans* (XP_050467798.1); S.p, *Schizosaccharomyces pombe* (NP_592848.1); S.c, *Saccharomyces cerevisiae* (CAA97095.1).

**Figure 4 jof-10-00177-f004:**
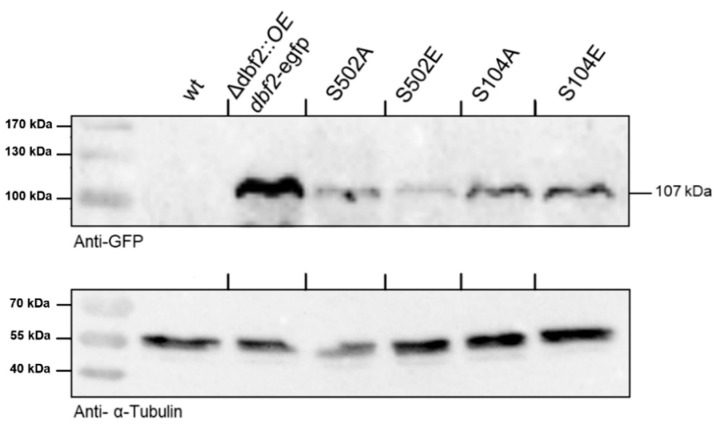
Expression of phospho-mutated variants of DBF2-GFP fusion proteins in strains as indicated. Strains were cultured on malt-cornmeal medium (BMM) as a surface culture for three days. Crude protein extracts, consisting of 10 μg of protein from each strain, were prepared and separated using SDS-PAGE. Western blot analysis was conducted using an anti-GFP antibody, with an anti-α-tubulin (55 kDa) antibody serving as a control. The DBF2-GFP fusion protein, with a molecular weight of 107 kDa, was detected in all phospho-mutant strains (S104A, S104E, S502A, S502E), as well as in the complemented strain. The wild type strain (wt) was used as a control.

**Figure 5 jof-10-00177-f005:**
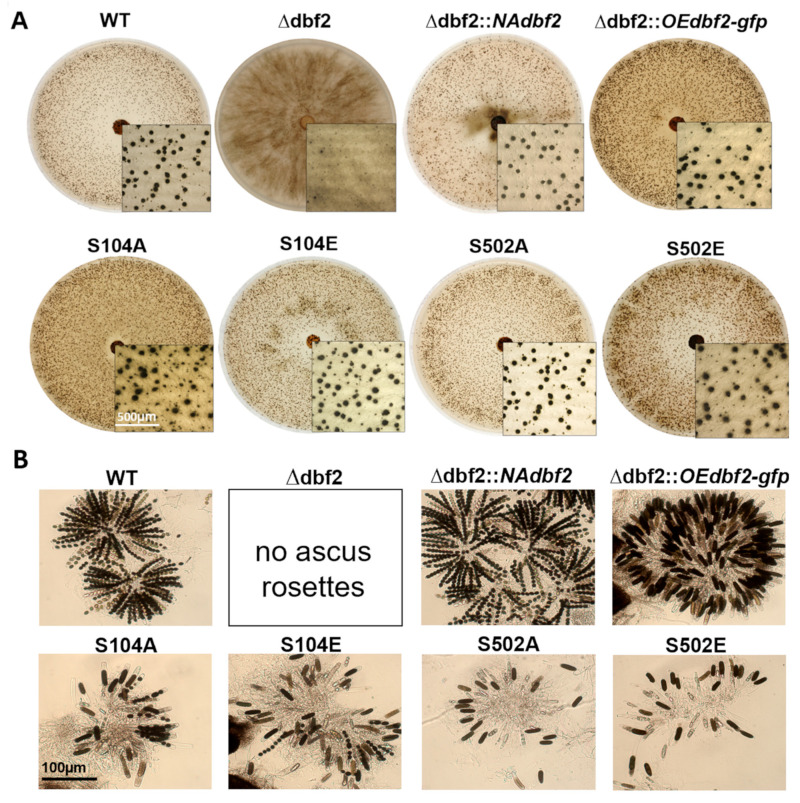
Sexual development and hyphal fusion in wild type, Δdbf2, complemented transformants, phospho-mimetic (S104E, S502E) and phospho-deficient dbf2 mutants (S104A, S502A). (**A**) Images of colony morphology and protoperithecia distribution were taken after incubation on BMM medium at 27 °C for seven days. Bar indicates 500 µm. (**B**) For microscopic analysis of perithecia and ascospores, strains were grown on BMM medium at 27 °C for 10 d. Bar indicates 100 µm. (**C**) For investigation of hyphal fusion, strains from (**A**) were grown on a layer of cellophane on MMS for two days. Investigation of hyphal fusion events (arrowheads) took place in a region 5–10 mm off the colony edges. Arrowheads mark hyphal anastomosis. Strains were grown on cellophane-coated MMS medium at 27 °C for 2–4 days. Scale bar is 20 µm.

**Figure 6 jof-10-00177-f006:**
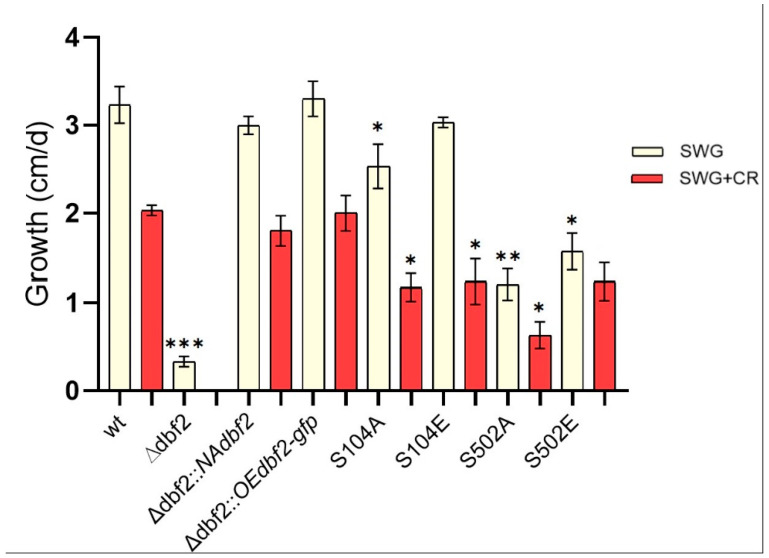
Vegetative growth and stress response of wild type, Δdbf2, complemented transformants and phospho-mimetic Δdbf2::OEdbf2-S104E, Δdbf2::OEdbf2-S502E (S104E, S502E), and phospho-deficient dbf2 mutants Δdbf2::OEdbf2-S104A, Δdbf2::OEdbf2-S502A (S104A, S502A). Vegetative growth and cell wall stress response of DBF2 kinase mutants. Sensitivity against Congo Red (0.01% (*v*/*v*) SDS plus 2 mg/mL CR) was monitored on petri dishes for 7 consecutive days. Shown are average growth rates on SWG (yellow bars) and SWG + CR (red bars) and standard deviations from three independent experiments are shown. Significant differences of growth length from that of the wild type are indicated by asterisks and were evaluated by a two-sided student’s *t*-test (***, *p* ≤ 0.001, **, *p* ≤ 0.01, *, *p* ≤ 0.05).

**Figure 7 jof-10-00177-f007:**
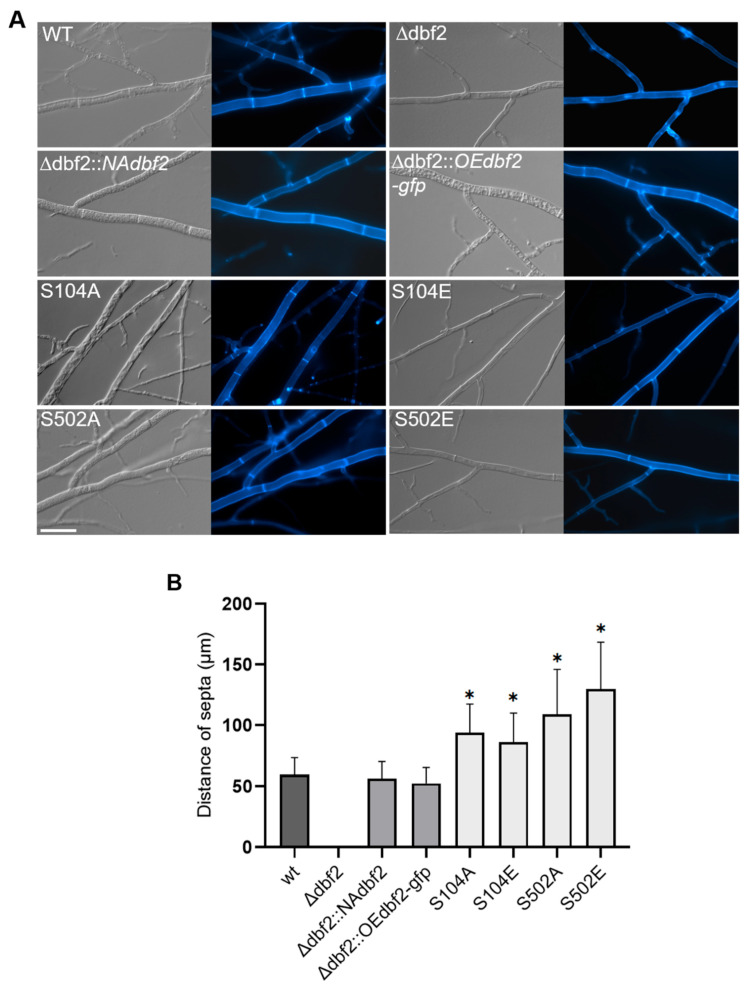
Septation phenotype of wild type, Δdbf2, complemented transformants and phospho-mimetic (S104E, S502E) and phospho-deficient dbf2 mutants (S104A, S502A). (**A**) Images of septation phenotypes from strains as indicated. Fluorescence microscopy was done with CFW stained mycelia. The scale bar corresponds to 20 μm (**B**) Quantitative data of septum distances. Significant differences of septum distances from that of the wild type and complemented strains are indicated by asterisks and were evaluated by a two-sided student’s *t*-test (*, *p* ≤ 0.01).

**Figure 8 jof-10-00177-f008:**
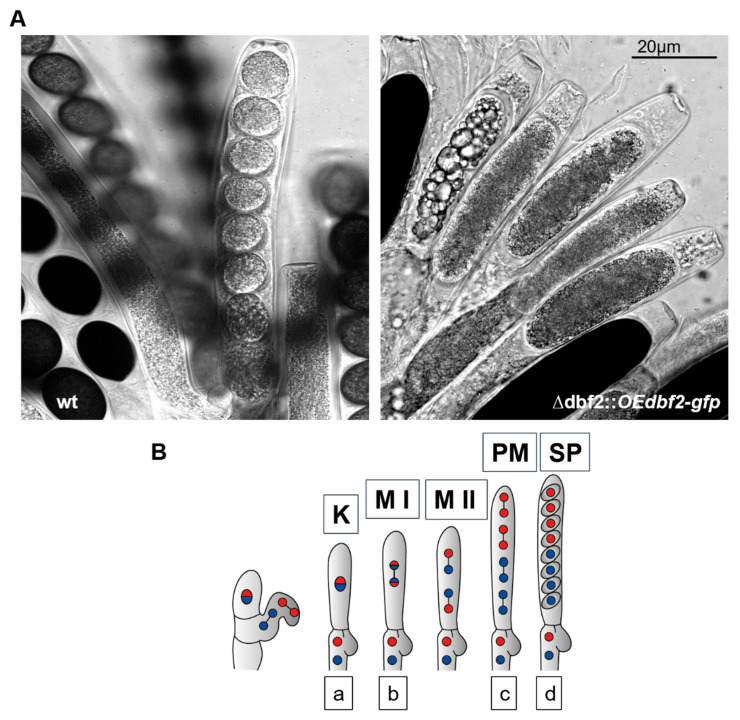
Nuclear division in asci from wild type and dbf2 overexpression strains. (**A**) Microscopic images show asci from wild type and dbf2 overexpression strains. In the latter, only large ascospores having a sausage-like shape are seen. (**B**) Schematic drawing of nuclear division during ascospore formation (Kück, unpublished). (**C**) DAPI staining to visualize nuclei in developing asci from wild type and dbf2 overexpression strains. a, b, c, and d indicate nuclear division stages, as shown in (**C**); scale bar is 20 µm.

## Data Availability

Plasmids and strains generated in this study are available upon request.

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
