# Peer review of "STRIPAK Dependent and Independent Phosphorylation of the SIN Kinase DBF2 Controls Fruiting Body Development and Cytokinesis during Septation and Ascospore Formation in Sordaria macrospora"

_jof, 2024, doi:10.3390/jof10030177_

Round 1
Reviewer 1 Report
All in all: great work that should be urgently published after minor revision!
The paper submitted stringently tests Dbf2 for function using a knock-out as well as phosphorylation (mimicking) mutants. This yields very good information, as the role of a regulation in phosphorylation (rather than fixed states of P) can be shown. The paper is interesting for both the fungal community and even beyond that. The involvement in sexual development of ascomycetes is detailed and very well corroborated.
There are a few minor comments:
1. l. 188 the dfb2 should be italics.
2. Please add somewhere that Sordaria is homothallic so as to make the readers aware that for sexual development no mate is required
3. The complementation with gpd promotor led to loss of ascospore formation. As the role of septation seems specifically important for hook formation and nuclear assortment prior to meiosis, could you discuss that fact further? Along that line for discussion: would dbf2 be present in basidiomycete genomes as well?
4. Could you state that the sausage-like super big ascospores contain 8 nuclei earlier? I found it specifically in the discussion.
5. May be you could discuss a pinpointing the point of action in cytokinesis and cell wall stress signaling? Like expression of cell wall synthesis genes? It would go along with the vegetative growth and septation phenotypes nicely…
Author Response
Reviewer #1
Major comments
All in all: great work that should be urgently published after minor revision!
Detail comments
The paper submitted stringently tests Dbf2 for function using a knock-out as well as phosphorylation (mimicking) mutants. This yields very good information, as the role of a regulation in phosphorylation (rather than fixed states of P) can be shown. The paper is interesting for both the fungal community and even beyond that. The involvement in sexual development of ascomycetes is detailed and very well corroborated.
Thank you for the positive response concerning our manuscript and for your advises to discuss some further aspects.
There are a few minor comments:
- l. 188 the dfb2 should be italics.. Was done
- Please add somewhere that Sordaria is homothallic so as to make the readers aware that for sexual development no mate is required
Was introduced on line 27 and 78
- The complementation with gpd promotor led to loss of ascospore formation. As the role of septation seems specifically important for hook formation and nuclear assortment prior to meiosis, could you discuss that fact further?
We now discuss the lack of septum formation being important for hook formation (line 406ff).
Along that line for discussion: would dbf2 be present in basidiomycete genomes as well?
As to the best of our knowledge, there are only unpublished data on basidiomycetes.
- Could you state that the sausage-like super big ascospores contain 8 nuclei earlier? I found it specifically in the discussion.
This has now been introduced on line 358
- May be you could discuss a pinpointing the point of action in cytokinesis and cell wall stress signaling? Like expression of cell wall synthesis genes? It would go along with the vegetative growth and septation phenotypes nicely…
We thank the reviewer for this comment and address the point at line 467 of the discussion.
Reviewer 2 Report
see detail comments
This paper studied the role of DBF2 during sexual development and vegetative growth in the fungus Sordaria macrospora. The paper is well written and presents solid evidence. Below are the comments.
1. In the abstract, the fungus could be briefly introducted and point the unsolved problem in the MS.
2. In introduction, how about the DBF2 orthologue in candida albicans? Is there any report?
3. Line 246 and 250, there should be reference.
4. From line 269, why S is not mutated to D?
5. Figure 6, the images of the plate may be shown.
6. Why figure 1 is not in color as in figure 7?
7. How about dapi staining in dbf2 deletion strain?
Author Response
Reviewer #2
Major comments
see detail comments
Detail comments
This paper studied the role of DBF2 during sexual development and vegetative growth in the fungus Sordaria macrospora. The paper is well written and presents solid evidence. Below are the comments.
The reviewer states that the paper is well written with solid data, thank you very much.
- In the abstract, the fungus could be briefly introducted and point the unsolved problem in the MS.
In the abstract, we mention now that the fungus is a homothallic ascomycete, line 27
- In introduction, how about the DBF2 orthologue in candida albicans? Is there any report?
We report now on lines 68,69 about the DBF2 homologue from C. albicans.
- Line 246 and 250, there should be reference.
The reviewer wishes to have the references for DBF2 homologues in other fungi. We have introduced the accession numbers for all homologous proteins in the legend of Fig. 3
- From line 269, why S is not mutated to D?
We are aware of this option. In our case, we have followed the strategy of related high-quality papers, such as Grallert et al. Nature 517, 94–98 (2015) https://doi.org/10.1038/nature14019; Conti et al. Nat Commun 14, 310 (2023). https://doi.org/10.1038/s41467-023-36035-9; Lechtenberg et al. Nat Commun 12, 7047 (2021). https://doi.org/10.1038/s41467-021-27343-z
- Figure 6, the images of the plate may be shown.
We refrain from showing at least 16 times two plates in Figure 6. This is not only an unproportional extension of our paper, but also it is not helpful for the understanding of the results. We hope the reviewer can except our decision.
- Why figure 1 is not in color as in figure 7?
We prefer to display fluorescence images in black and white for a better contrast of the DAPI staining. This was done in figures 1 and 8. In Figure 7, we show the fluorescence microscopy figures in blue to distinguish them clearly from the grayscale images obtained from light microscopy.
- How about dapi staining in dbf2 deletion strain?
This strain is sterile and does not produce any perithecia, asci or ascus spores. Therefore, we did not see the need for any DAPI staining
Reviewer 3 Report
This research addresses the function of septation initiation network kinase DBF2 in the reproduction of the model fungus Sordaria macrospora. The approach used gene knockout, complementation and the construction of strains with altered amino acids such to create versions that are phosphor-mimetic or -deficient. Depending on the alleles involved, a variety of impacts was observed related to vegetative septation and formation of ascospores during the sexual cycle. The work is comprehensive in the level of replication in each of the experiments performed. This research should appeal to people working on the genetics of fungal development. The authors also point how conserved these pathways are in eukaryotes, and thus of relevance beyond fungi.
A few minor typographical points are as follows:
L63: ‘In’ to ‘in’.
Line 165: add space ‘NP-40, 1 mM’.
Line 166: change ‘1,3’ to ‘1.3’.
Line 169: delete ‘.’.
Line 171: ‘Bio-Rad’ for consistency.
Line 190: change ‘xy’ to a number.
Line 217: remove italics from ‘and’.
Line 260-268: fix italics etc. On line 261 can remove the species abbreviations (e.g. (Sm)). For 266-268, better to use the style on the figure, so ‘Sm’ would be ‘S.m’ in italics.
In figure 3A, ‘572 aa’ can be slightly higher to match the space as above.
Line 337 and elsewhere: SDS vs CR? This was not clear in the methods, as they are written as if the two stresses were used separately, but in other places (354-355) it sounds like a mix was used. This could be clarified.
Line 456: missing a word ‘developmental genes/proteins/pathways/components identified’.
Line 558: italics on species name.
Line 668: remove italics for the antifungal.
Author Response
Reviewer #3
Major comments
This research addresses the function of septation initiation network kinase DBF2 in the reproduction of the model fungus Sordaria macrospora. The approach used gene knockout, complementation and the construction of strains with altered amino acids such to create versions that are phosphor-mimetic or -deficient. Depending on the alleles involved, a variety of impacts was observed related to vegetative septation and formation of ascospores during the sexual cycle. The work is comprehensive in the level of replication in each of the experiments performed. This research should appeal to people working on the genetics of fungal development. The authors also point how conserved these pathways are in eukaryotes, and thus of relevance beyond fungi.
We thank the reviewer for this very positive feedback.
Detail comments
A few minor typographical points are as follows:
We are thankful for the reviewer for these detailed corrections of the text, which all have been performed according to his advises.
L63: ‘In’ to ‘in’.
Line 165: add space ‘NP-40, 1 mM’.
Line 166: change ‘1,3’ to ‘1.3’.
Line 169: delete ‘.’.
Line 171: ‘Bio-Rad’ for consistency.
Line 190: change ‘xy’ to a number.
Line 217: remove italics from ‘and’.
Line 260-268: fix italics etc. On line 261 can remove the species abbreviations (e.g. (Sm)). For 266-268, better to use the style on the figure, so ‘Sm’ would be ‘S.m’ in italics.
In figure 3A, ‘572 aa’ can be slightly higher to match the space as above.
Line 337 and elsewhere: SDS vs CR? This was not clear in the methods, as they are written as if the two stresses were used separately, but in other places (354-355) it sounds like a mix was used. This could be clarified.
Line 456: missing a word ‘developmental genes/proteins/pathways/components identified’.
Line 558: italics on species name.
Line 668: remove italics for the antifungal.
Reviewer 4 Report
The manuscript entitled "STRIPAK dependent and independent phosphorylation of the SIN kinase DBF2 controls fruiting body development and cytokinesis during septation and ascospore formation in Sordaria macrospora" is well written and help to increase our understanding of the genes and pathways underlying key processes in fungi.
1. I think it will help the reader if a figure is added in the Introduction that provides more information about the pathways.
2. The authors should make sure that they provide the necessary information in the Materials and Method section. 2.a. For example, what and ho sequencing was done. Plasmid DNA sequencing was conducted either by Eurofins Genomics located in Ebersberg, Germany or by the Department for Biochemistry at Ruhr-University Bochum. (Lines-105 106).
2.b. "Using three technical replicates, growth and stress tests were conducted three times for strains, as mentioned in the results section." (Line 118)
3. Also, add the version and reference. For example, "The sequencing data obtained were subsequently analysed using SnapGene software." (Line 107).
4. Should this not be part of Materials and Methods, rather than Results: "For a functional characterization of DBF2, a dbf2 deletion strain (Δdbf2) was generated using a BsaI-mediated Golden Gate cloning system [37] (Suppl. Fig. S1)." (Lines 187 ).
5. The authors need to provide results in the Results section. e.g., "Our recent comprehensive global proteome and phosphoproteome analysis [13,49] provided valuable insights into potential dephosphorylation targets of STRIPAK." (Lines 239).
6. The authors should focus on results generated in this study and compare with other studies. For example, they should nog start a paragraph "According to a previous analysis, the de-phosphorylation of SmKIN3," (Line 469).
Author Response
Reviewer #4
The manuscript entitled "STRIPAK dependent and independent phosphorylation of the SIN kinase DBF2 controls fruiting body development and cytokinesis during septation and ascospore formation in Sordaria macrospora" is well written and help to increase our understanding of the genes and pathways underlying key processes in fungi.
We thank the reviewer for his positive response concerning our manuscript.
- I think it will help the reader if a figure is added in the Introduction that provides more information about the pathways.
We thank the reviewer for this suggestion. To provide a concise introduction, we have given several reviews and the paper by Stein et al. 2021 [reference 13] provides a scheme that is helpful for understanding the pathways. This paper has now been mentioned in the introduction.
- The authors should make sure that they provide the necessary information in the Materials and Method section.
2.a. For example, what and how sequencing was done. Plasmid DNA sequencing was conducted either by Eurofins Genomics located in Ebersberg, Germany or by the Department for Biochemistry at Ruhr-University Bochum. (Lines-105 106). Was done on line 120
2.b. "Using three technical replicates, growth and stress tests were conducted three times for strains, as mentioned in the results section." (Line 118) confirmed
- Also, add the version and reference. For example, "The sequencing data obtained were subsequently analysed using SnapGene software." (Line 107). Was done on line 121
- Should this not be part of Materials and Methods, rather than Results: "For a functional characterization of DBF2, a dbf2 deletion strain (Δdbf2) was generated using a BsaI-mediated Golden Gate cloning system [37] (Suppl. Fig. S1)." (Lines 187 ). Was moved to the M+M section line 116
- The authors need to provide results in the Results section. e.g., "Our recent comprehensive global proteome and phosphoproteome analysis [13,49] provided valuable insights into potential dephosphorylation targets of STRIPAK." (Lines 239).
In the sentences to follow, the results are described, for example the phosphorylation sites of DBF2. Beside that we are not allowed to present data again, which were already published by Märker et al. 2020, or by Stein et al. 2021.
- The authors should focus on results generated in this study and compare with other studies. For example, they should not start a paragraph "According to a previous analysis, the de-phosphorylation of SmKIN3," (Line 469).
In the forgone paragraph, we already discuss the results of this contribution, namely the phosphorylation of S104 and S502 sites in DBF2. This is followed by the sentence “According to a previous analysis….” To make this more obvious, we have change the formatting of this paragraph.